# Recent Advances in Stem Cell Differentiation Control Using Drug Delivery Systems Based on Porous Functional Materials

**DOI:** 10.3390/jfb14090483

**Published:** 2023-09-20

**Authors:** Yun-Sik Eom, Joon-Ha Park, Tae-Hyung Kim

**Affiliations:** School of Integrative Engineering, Chung-Ang University, 84 Heukseuk-ro, Dongjak-gu, Seoul 06974, Republic of Korea; eys0521@cau.ac.kr (Y.-S.E.); joonha95@cau.ac.kr (J.-H.P.)

**Keywords:** porous particle, stem cell, differentiation control, drug delivery system

## Abstract

The unique characteristics of stem cells, which include self-renewal and differentiation into specific cell types, have paved the way for the development of various biomedical applications such as stem cell therapy, disease modelling, and drug screening. The establishment of effective stem cell differentiation techniques is essential for the effective application of stem cells for various purposes. Ongoing research has sought to induce stem cell differentiation using diverse differentiation factors, including chemicals, proteins, and integrin expression. These differentiation factors play a pivotal role in a variety of applications. However, it is equally essential to acknowledge the potential hazards of uncontrolled differentiation. For example, uncontrolled differentiation can give rise to undesirable consequences, including cancerous mutations and stem cell death. Therefore, the development of innovative methods to control stem cell differentiation is crucial. In this review, we discuss recent research cases that have effectively utilised porous functional material-based drug delivery systems to regulate stem cell differentiation. Due to their unique substrate properties, drug delivery systems based on porous functional materials effectively induce stem cell differentiation through the steady release of differentiation factors. These ground-breaking techniques hold considerable promise for guiding and controlling the fate of stem cells for a wide range of biomedical applications, including stem cell therapy, disease modelling, and drug screening.

## 1. Introduction

Stem cells possess the unique ability for self-renewal and differentiation into distinct tissue cells [1,2,3]. This capability has driven significant advancements in tissue engineering and regenerative medicine [4,5,6,7,8]. Understanding the process of stem cell differentiation has enabled the creation of disease models, thus facilitating a deeper grasp of challenging diseases and the exploration of treatment methods [9,10,11,12]. Moreover, stem cells offer a promising basis for the development of specific organ and tissue models for drug screening [13,14,15]. Stem cells can also be used for promoting differentiation or transplantation to treat various health disorders [16,17,18] (Figure 1A).

The differentiation of stem cells into specific tissues and organs is determined by epigenetic regulation [19,20]. The differentiation process entails the activation of various signalling pathways, such as the Wnt signalling pathway. Therefore, research has been actively conducted to develop methods for the induction of specific signalling pathways [21,22,23,24,25,26,27,28]. A commonly employed approach involves culturing stem cells in differentiation media containing proteins and chemicals that stimulate the expression of genes associated with differentiation. This method is widely used to induce signalling pathways and stimulate stem cell differentiation [29,30,31,32,33]. Another strategy involves creating an extracellular matrix (ECM) environment for the expression of specific integrins in stem cells, thus effectively regulating differentiation. This approach aims to control differentiation by providing an ECM environment that mimics the natural conditions required for specific cell types to differentiate [34,35,36,37].

However, applying stem cell differentiation without considering the interplay between various differentiation factors and stem cells can pose several challenges. Firstly, stem cells may differentiate into undesired cell types or fail to exhibit proper cellular functions [38,39]. Furthermore, signalling pathways can also serve as catalysts for stem cells to undergo mutations and transform into cancer cells, depending on their regulation [40,41,42]. For instance, Wnt signalling is activated during the differentiation of various stem cells [43,44,45]. However, unregulated Wnt signalling can significantly contribute to various types of cancer [40,46,47,48,49,50] (Figure 1B). Moreover, high concentrations of differentiation factors in the differentiation medium impair cellular homeostasis, inducing stress in the cells and ultimately leading to apoptosis [51,52,53,54,55]. Additionally, differentiation factors are consumed and degraded by cells during the differentiation process, resulting in scarcity within 48 h of culture [56,57,58]. Therefore, novel techniques are needed to precisely control stem cell differentiation and induce them to differentiate into specific tissue cells.

The use of porous functional materials in drug delivery systems (DDS) is emerging as an innovative approach to controlling stem cell differentiation (Figure 1C). DDS involves the targeted delivery and release of drugs intended for specific diseases [59,60,61,62]. A variety of materials, including silica, polymer nanoparticles, metals/metal oxides, liposomes, and exosomes, have been employed as carriers for drug delivery [60,63,64]. Drug carriers based on porous functional materials offer several advantages, such as drug loading and release capabilities, due to their porous surface and internal empty space [65,66,67,68,69,70,71,72]. Additionally, porous materials are biocompatible and protect drugs from degradation [73,74,75]. These carriers can also be synthesised to align with the specific requirements of the target drug, including factors such as internal space and pore size [76,77,78]. The drug release rate can also be controlled through chemical and protein modifications of the porous surface [79,80,81,82]. Additionally, adhesion can be enhanced by incorporating ECM components such as Arg-Gly-Asp (RGD), fibronectin (FN) surface coating, and scaffolds embedded with porous materials [83,84,85,86]. This innovative approach enables the precise control of stem cell differentiation towards specific tissue types.

This review summarises the recent advancements in stem cell differentiation control based on drug delivery technologies using various types of porous functional materials that enable the regulation of stem cell differentiation, including osteogenesis, neurogenesis, and cartilage differentiation. Particularly, this review will focus on (i) mesoporous silica nanoparticles, (ii) polylactic-co-glycolic acid, (iii) metal organic frameworks, (iv) magnetic nanoparticles, and (v) upconversion nanoparticles (Table 1) (Figure 1D).

## 2. Mesoporous Silica Nanoparticle

Mesoporous silica nanoparticles (MSNs) are synthetic materials based on silica precursors such as tetraethyl orthosilicate (TEOS) [102,103,104]. MSN has garnered significant attention in the field of DDS due to its notable chemical stability, surface functionality, biocompatibility, tuneable pore size, and remarkable drug loading capacity [105,106,107,108]. Therefore, previous studies have sought to control stem cell differentiation by loading differentiation factors into MSNs.

In 2019, successful osteogenesis of human mesenchymal stem cells (hMSCs) was achieved using spherical, 80-nm-sized MSNs loaded with dexamethasone (DEX) [87]. The MSNs were then spin-coated onto glass coverslips to form a film (Figure 2A). The release of DEX was assessed through the calcein model, revealing that 97.0 ± 2.6% of hMSCs had absorbed calcein after one day of culture (Figure 2B). Additionally, hMSCs were cultured on the MSN film for 7 days to demonstrate the ability of the film to support cell adhesion and biocompatibility. The hMSCs were allowed to differentiate for 28 days, during which the expression of the osteogenic factor alkaline phosphatase (ALP) and matrix mineralisation were quantified. Differentiation was performed using 100% normal media exchange (negative control), MSNs without DEX + 100% normal media exchange, DEX-loaded MSN film + 100% and 50% normal media exchange, DEX-loaded MSN film + 100% osteogenic media exchange, and 100% osteogenic media exchange (positive control). On day 28, the 50% normal media exchange group exhibited the highest values (Figure 2C). The study demonstrated that MSN film enhances osteogenesis in MSCs. However, the influence of substrate properties and the controlled release of differentiation factors on the stem cell differentiation process could not be distinguished.

In 2019, osteogenesis and angiogenesis were successfully induced in bone marrow-derived mesenchymal stem cells (bMSCs) using MSNs capable of releasing DEX and QK peptides [88]. Spherical 170 nm MSNs were synthesised, and their surfaces were modified with chitosan. DEX was then loaded into the synthesised MSN, and the surface of the MSNs was modified with QK peptide, mimicking vascular endothelial growth factor (VEGF). DEX exhibited release rates of 82% and 61% after 21 days at pH 6.0 and 7.4, respectively. Moreover, 60% of the QK peptide was released at 48 h, whereas 81% was released on day 7. The chitosan-modified MSN exhibited excellent biocompatibility throughout 48 h of bMSC culture. The osteogenesis of bMSCs was confirmed through ALP activity, expression of osteogenic genes and proteins, and in vitro mineralisation. On day 14 of culture, the ALP activity, osteogenic gene expression, osteogenic protein levels, and in vitro mineralisation showed significantly higher values compared to the control group. The induction of angiogenesis was evidenced by vascular density characterisation. The group exposed to the released QK peptide exhibited higher vascular density compared to the negative control group and showed a similar vascular density as VEGF. These results demonstrate the suitability of MSNs releasing DEX and QK peptides for bMSC differentiation and angiogenesis induction.

MSNs can also be incorporated into scaffold structures to control stem cell differentiation. In a study by Zhou et al., a chitosan-alginate-gelatin (CAG) -based porous scaffold incorporated with MSN loaded with differentiation factors was developed [89]. The aim was to successfully induce osteogenesis in rat bone marrow-derived mesenchymal stem cells (rbMSCs). The MSNs were synthesised as spherical particles with a size of 210.6 nm. Dexamethasone (DEX) was loaded into the MSNs, which were then incorporated into calcium phosphate crystals deposited on the CAG scaffold (pcCAG). The release of DEX was monitored over a 28-day period, revealing a consistent rise in cumulative release, with 65.7 ± 4.3% of the loaded DEX released by day 28. The biocompatibility of the MSN@pcCAG scaffold was demonstrated by culturing rbMSCs on the scaffold for 5 days. The osteogenesis of rbMSCs was assessed through ALP expression. After 14 days of culture, the MSN@pcCAG scaffold exhibited significantly higher levels of ALP expression, indicating successful osteogenesis. Furthermore, substantial bone regeneration was observed when the MSN@pcCAG scaffold was introduced into a rat calvarial bone defect model for 8 weeks. These findings demonstrate that scaffolds incorporating DEX-releasing MSNs are well-suited for stimulating osteogenesis both in vitro and in vivo.

The porous structure of MSNs has been found to be conducive to the release of differentiation factors while also promoting cell adhesion. MSNs can be applied in various formats, such as particles, films, and scaffolds, thus allowing for tailored control of stem cell differentiation. These findings demonstrate that the use of MSN for stem cell differentiation control extends beyond osteogenesis and holds value for controlling differentiation in various stem cell types.

## 3. Polylactic-Co-Glycolide Acid

Polymeric materials play a pivotal role in topical drug delivery, with polymers such as poly(lactic-co-glycolic acid) (PLGA) and polyethylene glycol (PEG) being prominent examples [109]. Notably, PLGA offers the advantage of flexible control over drug release rates based on the polyglycolic acid (PGA) to polylactic acid (PLA) ratio [110,111,112,113]. In turn, this enables the design of differentiation platforms tailored to the distinct differentiation durations of various stem cells.

Tissue engineering scaffolds must provide a conducive matrix for treating or regenerating tissues while maintaining mechanical stability [114]. To achieve this, hydrogels with physical properties akin to biological tissue, achieved through water content regulation, have been recently developed [115]. Particularly, gelatin has been found to modulate cell adhesion, proliferation, and differentiation, making it a fitting scaffold material for cartilage tissue engineering [114,116]. In 2022, a DDS that can successfully promote cartilage formation was reported, which involved a complex of transforming growth factor beta-1 (TGF-β1) loaded into gelatin/PLGA-PEG-PLGA nanoparticles [90]. The fabricated PLGA nanoparticles, measuring approximately 80–100 nm, were loaded onto and solidified within gelatin. To create an efficient DDS for cells, a porous structure measuring 107.09 ± 55.11 μm was introduced into the gelatin/PLGA nanoparticle scaffold, thus creating a 3-dimensional (3D) environment conducive to nutrient transport, oxygen diffusion, cell attachment, growth, and migration. Human dental pulp-derived mesenchymal stem cells (h-DPSCs) were seeded onto this scaffold, where the gelatin within the scaffold facilitated cell adhesion, viability, and proliferation due to its unique water absorption properties. TGF-β1 was rapidly released from the PLGA nanoparticles and scaffolds during the first 9 days, with 75–78% of the total loaded amount being released by the 21st day. Biocompatibility was demonstrated by culturing h-DPSCs on a scaffold for 14 days, at which point the formation of an expanded cytoskeleton of chondrocytes was successfully confirmed. After 21 days, differentiation into a chondrocyte phenotype was further demonstrated. However, the study was unable to decouple the effects of substrate properties and the controlled release of differentiation factors on the stem cell differentiation process. These results suggest that the biocompatible gelatin/PLGA nanoparticle scaffold containing TGF-β1 can successfully differentiate into chondrocytes by mimicking a 3D environment suitable for adhesion, proliferation, and differentiation of h-DPSCs.

The applications of alginate-based bioink for 3D bioprinting have remained limited due to printing defects and structural instability [117]. To overcome these limitations, a recent study developed an alginate scaffold for inducing osteogenesis in mesenchymal stem cells (MSC) using PLGA as a bioink [91]. Bone morphogenetic protein-2 (BMP-2) was loaded onto PLGA nanoparticles generated through a water-in-oil-in-water double emulsion, which was used for bone differentiation of MSCs. The fabricated PLGA nanoparticles were spherical, with a size of 172 ± 56 nm. Moreover, their surface charge was negative to enhance the structural stability and printability of the printed structures. Approximately 72% of the loaded BMP-2 was released from the PLGA scaffold over the course of 2 weeks. MSCs were cultured on the scaffold for 7 days, thus demonstrating the biocompatibility of the proposed material. Furthermore, robust osteogenic activity was demonstrated through gene expression analysis of MSCs. However, the study was unable to disentangle the influences of substrate characteristics from the controlled release of differentiation factors on the stem cell differentiation process. Collectively, these findings underscore the feasibility of the generated PLGA/alginate scaffold as a 3D printable structure with clinical applicability in the field of bone tissue engineering.

Achieving high mechanical strength is a critical challenge in the design of scaffolds intended for bone implantation. Additionally, the achievement of a physical form akin to the ECM and ensuring biocompatibility without posing physiological concerns are also crucial. To address these issues, Qasim et al. developed a nanohybrid scaffold by embedding PLGA microparticles (PLGA MPs) into biodegradable polycaprolactone (PCL) nanofibers [92]. This nanohybrid scaffold was loaded with TGF-β3 to facilitate the cartilage differentiation of human mesenchymal stem cells (hMSCs) (Figure 3A). The synthesised PLGA MPs were spherical with a 10–50 μm size range and a surface porosity of 1.5–10 μm (Figure 3B). The loading efficiency of this novel material exceeded 80%, with 91% being released after 14 days of culture. After seeding, the proliferation of hMSC on the manufactured nanohybrid scaffold was continuously monitored for 14 days (Figure 3C). However, the study was unable to differentiate between the influence of substrate properties and the controlled release of differentiation factors on the stem cell differentiation process. Nevertheless, intracellular glycosaminoglycan (GAG) content was quantified, and a successful elevation of GAG levels was observed as differentiation progressed. Finally, the fabricated PLGA MP-PCL nanohybrid scaffold successfully enabled the sustained release of differentiation factors, thus confirming its capacity to facilitate hMSC proliferation and differentiation in vitro. These results demonstrate the viability of scaffolds loaded with PLGA nanoparticles for safeguarding unstable TGF-β3 and promoting hMSC differentiation.

By leveraging the structural versatility of PLGA based on the PGA-PLA ratio, the design of scaffold structures can be tailored to cater to the unique differentiation of various stem cell types by controlling the rate of the biodegradation reaction. Moreover, by introducing additives with varying biocompatibility, the stability and mechanical strength of scaffold bioprinting can be enhanced. Therefore, it is important to confirm the findings of these studies in vivo, as this would enable the identification of novel functional candidates with potential applications in the field of stem cell therapy through ex vivo regeneration.

## 4. Metal-Organic Frameworks

Metal-organic frameworks (MOFs) constitute a category of porous coordination polymers composed of metal nodes and organic linkers [77,118,119]. Importantly, MOFs exhibit a variety of unique advantages, including high surface area and porosity, chemical and thermal stability, tuneable pore size, internal drug protection, and easy chemical functionalisation [120,121,122]. These attributes position MOFs as outstanding candidates for the control of stem cell differentiation.

One of the major challenges encountered in the differentiation of neural stem cells (NSCs) is the rapid degradation of retinoic acid (RA), a pivotal differentiation factor. Sustaining a consistent RA concentration requires not only the development of materials capable of controlled RA release but also the protection of the released RA from degradation. To achieve this, a study reported the successful differentiation of NSCs using a nanohole array based on UiO-67 (UiO: University of Oslo) MOF, which releases RA (Figure 4A) [93]. The synthesised UiO-67 was 177 nm in diameter and was based on zirconium (Zr) and 4,4′-biphenyldicarboxylate (Figure 4B). After loading RA onto UIO-67 and making a nanohole pattern using laser interference lithography on indium tin oxide (ITO) glass, RA was successfully encapsulated within the nanoholes. The reduction of RA degradation was verified by subjecting the RA-loaded UiO-67 to sunlight. RA release was monitored for a total of 26 days, with 6.6 × 10^−5^ mM/mg being released per day for the first 2 days, after which the release rate stabilised at 0.7 × 10^−5^ mM/mg per day for the remaining 24 days (Figure 4C). Biocompatibility analysis results revealed no substantial differences in the levels of the Nestin and sex-determining region Y-box 2 (SOX2) NSC markers when compared to the control group (bare ITO). The NSC differentiation process was allowed to continue for 14 days. By day 10 of culture, discernible variations in marker expression for mature neurons were evident. Comparison of the expression of representative neuronal markers NeuroD1 and microtubule-associated protein 2 (MAP2) revealed a 9.54-fold increase in NeuroD1 and an 8.75-fold increase in MAP2 compared to the control group. Moreover, the expression of messenger RNA (mRNA) was 43.7-fold higher for NeuroD1 and 41.3-fold higher for MAP2 (Figure 4D). This study thus demonstrated that the UiO-67-confined nanoarray effectively shielded RA from degradation, thereby promoting NSC differentiation. This suggests that protecting differentiation factors in vivo by encasing them in UiO-67 can greatly enhance the effectiveness of stem cell therapy.

One of the main challenges of regenerative medicine is the limited presence of MSCs in vivo, thus requiring the direct targeting of differentiation factors for effective tissue repair. To overcome these challenges, a membrane-coated zeolitic imidazolate framework-8 (ZIF-8) was developed. Successful osteogenesis induction was achieved by encapsulating ZIF-8, which releases dexamethasone (DEX) [94]. ZIF-8 was synthesised using zinc (Zn) and 2-methylimidazole, resulting in 100-nanometer particles. DEX was loaded into the synthesised ZIF-8 and then encapsulated with the stem cell membrane obtained from MSCs. DEX was then released over a total of 24 days, with a loading efficiency of 78% and a release rate of 0.624 mg per particle. The biocompatibility of various concentrations of ZIF-8 was demonstrated by culturing MSCs for 48 h. Lower concentrations exhibited no cytotoxicity, whereas cell viability decreased to 30% at 50 and 100 µg/mL. MSC osteogenesis through membrane-encapsulated ZIF-8 was evaluated over a 15-day culture period. Notable differences in ALP expression compared to the control group were evident, which were accompanied by a significant increase in osteopontin (OPN) and osteocalcin (OCN) mRNA expression. Additionally, significant results were observed in a rat femur defect model for bone repair. These results provide promising evidence for in vivo MSC-specific delivery of DEX to induce effective osteogenesis using ZIF-8 coated with MSC membranes.

Reactive oxygen species (ROS), generated from the metabolic activity of cells, are known to inhibit osteogenesis [123]. A study published in 2020 reported a MOF capable of releasing both differentiation factors and antioxidants through its reaction with H_2_O_2_, a representative ROS. Materials from the Institut Lavoisier-100(Fe) (MIL-100(Fe)), which release small interfering RNA (siSOX9) and RA, were used to induce neuronal differentiation [95]. MIL-100(Fe) with a 100–200 nm size range was synthesised using Fe and trimesic acid (H3BTC). Upon cellular uptake, MIL-100(Fe) is decomposed due to the presence of H_2_O_2_ within cells. Ceria nanoparticles were thus loaded concurrently as antioxidants to counteract the effects of H_2_O_2_. RA and siSOX9 were loaded on MIL-100 (Fe). The release of RA was measured at various H_2_O_2_ concentrations. At 1000 μM H_2_O_2_, 40% of RA was released within 5 h. Biocompatibility was evaluated by culturing the cells for 24 h in various concentrations of MIL-100 (Fe). Moreover, MIF-100 (Fe) and NSCs were co-cultured for 14 days, and the degree of differentiation was measured. The expression of the SOX9 gene decreased compared to the conventional method, demonstrating the regulation of SOX9 expression by siSOX9. Upon comparing the staining of neuron marker Tuj1 and glial cell marker glial fibrillary acidic protein (GFAP), the control group primarily differentiated into glial cells, whereas the MIL-100(Fe) treatment group favoured neuronal differentiation. These results demonstrate the ability of the developed MIL-100(Fe) material to reduce the ROS generated by metabolic activity and NSC differentiation. In turn, this suggests that MIL-100(Fe) can enhance various functions in addition to stem cell differentiation.

MOFs, which can be synthesised from various materials, enable size control based on the combination of linkers and metal nodes. Thus, co-loading particles with two or more differentiation factors and supplementary functions can further enhance stem cell differentiation efficiency. Taken together, these findings demonstrate that MOFs can enable the release of multiple differentiation factors, thus allowing for more precise differentiation control compared to conventional methods. Therefore, the application scope of these materials could be extended to ex vivo regeneration.

## 5. Magnetic Nanoparticles

The fields of regenerative medicine and tissue engineering have recently embraced the use of magnetic nanoparticles as an innovative strategy for spatially deploying and stimulating stem cells [124,125,126]. Previous studies have harnessed magnetic force dynamics to drive the differentiation of stem cells into cartilage, fat, and bone tissues [127,128,129]. Notably, magnetic nanoparticles internalised within the platform and scaffold where stem cells are cultured can exhibit biological activity, contributing to processes such as cell targeting, drug delivery, and pathway stimulation [130,131,132]. These properties make magnetic nanoparticles promising materials for actively inducing stem cell differentiation, thereby providing an additional advantage for various regenerative medicine applications.

Distraction osteogenesis is a widely employed technique for bone defect repair [133]. However, the clinical applications of this technique have remained limited [134]. To overcome this issue, approaches involving physical stimulation and the injection of diverse biological factors are under investigation. Particularly, the mechanisms through which magnetic mesoporous silica-coated nanoparticles stimulate the release of stem cell growth factors have recently garnered increasing attention. A recent study reported the development of magnetic nanoparticles encapsulated with mesoporous silica and the successful differentiation of bone cells by releasing Si ions in response to stimulation (Figure 5A) [96]. The synthesised magnetic nanoparticles were 55 ± 16 nm in size, as demonstrated by TEM analyses. The core that endowed the nanoparticles with their magnetic properties was an iron oxide nanocrystal, which was encapsulated within a silica shell. Live/dead assays using the fabricated nanoparticles indicated excellent biocompatibility, as there were no discernible toxic effects in comparison to non-treated groups. Furthermore, the nanoparticle-treated group exhibited increased expression patterns of Axin2, c-myc, and β-catenin relative to the control group. These results collectively underscore the ability of the proposed material to promote stem cell differentiation (Figure 5B). Finally, treating mice with nanoparticles yielded no fatalities or side effects, and after 4 weeks, more newly formed bone and cartilage tissues were successfully generated in the treated groups compared to the control group (Figure 5C). These findings challenge the previously held notion that magnetic nanoparticles are highly toxic, instead demonstrating elevated biocompatibility. Collectively, these results suggest that magnetic nanoparticles can potentially enhance osteogenesis in both in vivo and clinical applications.

3D platforms have been recognised for their potential to enhance bone regrowth by providing a temporary framework that offers a conducive environment for cell attachment and growth. However, targeting loaded nanoparticles to bone cells with this design presents challenges, and studies on the potential contamination of bone cells by external factors are scarce. To address this issue, a multi-layered 3D platform loaded with drugs and multiple bioactive nanoparticles was developed to promote bone tissue repair and regrowth [97]. Nano-sized iron oxide-based magnetic nanoparticles were synthesised using a co-precipitation method in a hypoxic environment to minimise oxidation and characterised to have a diameter of 15 ± 3 nm through SEM and TEM images. The scaffold forming the body of the 3D platform was constructed using PLA via 3D printing technology, incorporating a porous structure through the complex loading of collagen, HA, minocycline, and magnetic nanoparticles. The fabricated magnetic 3D scaffolds demonstrated the ability to inhibit *Staphylococcus aureus* growth due to their porous structure, which facilitated antibiotic delivery. Collagen in the scaffold enabled effective minocycline release, preventing initial bacterial attachment and reducing contamination. The long-term retention of magnetic properties in the fabricated 3D scaffold was evaluated using human bone marrow mesenchymal stem cell (hBMSC) lines. Approximately 80% of the total amount of minocycline was released within 24 h, with hBMSC viability decreasing after 2 weeks and recovering after 3 weeks. The proliferation and differentiation of hBMSCs cultured on the 3D scaffold were also monitored. From day 15, organised cell monolayers of elongated and flat cells appeared extensively, indicating the successful formation and expansion of filopodia in hBMSCs. This study demonstrated exceptional bioactivity in promoting bone cell differentiation while simultaneously preserving the magnetic properties of iron oxide and guarding against *S. aureus* contamination. This suggests that this approach could reliably prevent external contamination that may occur in the ex vivo cell differentiation stage prior to transplant surgery.

Recently, highly biocompatible artificial materials produced by 3D printing technology are being used for biomedical purposes. However, studies have indicated that artificial scaffolds made of single materials exhibit a variety of disadvantages, such as inflammation, low strength, and low bioactivity. To address these limitations, a recent study developed a graphene oxide and magnetic nanoparticle complex that promoted bone regeneration and tumour treatment [98]. The use of hydroxyapatite (HA) and sodium alginate (SA) in this research led to a swift and robust hardening process, forming chemical cross-links that enveloped the graphene oxide particles with polymer chains. Notably, the resulting scaffold exhibited mechanical stability as the graphene oxide remained embedded beneath the surface. Additionally, the proposed material exhibited a pore size of 300 μm, which is favourable for cell growth and nutrient transport, as characterised by SEM. First, the authors tested the antitumor potential of the proposed material by generating an alternating magnetic field (AMF) on the fabricated graphene oxide-loaded magnetic scaffold. The death of cancer cells was attributed to the heat generated by the alternating magnetic field, which was stored due to the heat-resistance properties of graphene oxide. Afterwards, the fabricated scaffold loaded with graphene oxide was utilised for an in vitro osteogenesis assessment involving bone marrow-derived mesenchymal stem cells (BMSCs) over a 3-week period. The outcomes of this study revealed an upregulation in the expression of genes associated with bone differentiation, including BMP-2, OCN, OPC, and RUNX-2, demonstrating successful bone differentiation. Notably, the study demonstrated the uniform distribution of Fe_3_O_4_ particles within graphene oxide, highlighting the potential of this scaffold as an antitumor platform while also showcasing its stimulatory effects on mouse BMSC differentiation. These findings thus highlight the potential applicability of this approach for clinical applications.

In general, although magnetic nanoparticles are commonly employed for cell differentiation via drug release triggered by magnetic stimulation [135], previous studies have also demonstrated that the magnetic nanoparticles discussed in this review offer additional benefits, including antitumor effects and the prevention of external contamination. These studies provide insights into the feasibility of commercialising these materials for future transplantation surgeries, as this technology applies not only to bone-related applications but also to the differentiation of stem cells into cartilage, adipose tissue, and nerve cells.

## 6. Upconversion Nanoparticle

Upconversion nanoparticles (UCNPs) are a class of optical nanomaterials doped with lanthanide ions [136,137,138]. Lanthanide ions have efficient electron transitions in their 4f electron shells, enabling the conversion of multiple low-energy photons into a single high-energy photon through upconversion [139,140,141]. UCNPs for drug delivery are typically synthesised based on a core-shell structure [142,143]. Core-shell UCNPs generate spaces for drug loading, typically through a mesoporous silica coating [143,144,145,146]. The synthesised UCNPs absorb near-infrared (NIR) light at 808 or 980 nm and subsequently emit photons of the desired wavelength. The incorporation of photoresponsive polymers on the particle surface, employing mechanisms like photocleavage and photoswitching, provides an effective means to control drug release. This approach offers the advantage of directly controlling the initiation of drug release [147,148,149,150].

In 2022, successful induction of MSC osteogenesis was achieved by synthesizing UCNPs that release icariin (ICA) (Figure 6A) [99]. Core-shell UCNPs doped with thulium (Tm)/erbium (Er) were synthesised with a size of 40 nm. The synthesised UCNPs were coated with mesoporous silica and conjugated with a photocaged linker 4- (hydroxymethyl) -3-nitrobenzoic acid, cap material β-cyclodextrin (CD), and Arg-Gly-Asp (RGD) for adhesion, resulting in 60-nm-sized particles. ICA was then loaded into the UCNP. When irradiated with 980 nm NIR, UV light was emitted, cleaving the 4- (hydroxymethyl) -3-nitrobenzoic acid and releasing ICA. The UCNP without NIR irradiation exhibited 11.28% release, whereas samples irradiated with an intensity of 1 and 2 W/cm^2^ for 1 h exhibited release rates of 62.07% and 78.13%, respectively. Furthermore, at an intensity of 1 W/cm^2^, the authors reported release percentages of 33.56% at 0.5 h, 65.54% at 1 h, and 76.96% at 2 h (Figure 6B). Upon cultivating MSCs with varying UCNP concentrations (0, 50, 100, 500, and 1000 μg/mL) over a 24-h period, no cytotoxicity was observed. When ICA-loaded UCNP was used for differentiation, the expression levels of Runx2, BMP-2, and osteopontin (OPN) were more than twice as high in the ICA-loaded UCNP + NIR group compared to the control group (Figure 6C). Significant differences in ALP and Alizarin red (ARS) staining compared to other groups were also observed (Figure 6D). Additionally, successful in vivo treatment was achieved by applying the developed UCNP to an osteoporosis rat model. The UCNPs were injected into the rat joints of an osteoporosis model, and ICA was released through NIR irradiation. This study showcased the controlled release of UCNP through modulation of NIR intensity and duration, highlighting its efficacy for osteoporosis treatment through micro-CT, Masson’s trichrome analysis, immunohistochemical staining, and the quantification of OCN and ALP expression. Collectively, these findings highlight the promising applicability of UCNP in regenerative medicine applications.

Stem cell therapy stands as a pivotal technology for addressing a wide range of diseases. For example, successful chondrogenic differentiation of MSCs was accomplished through the utilisation of UCNPs capable of releasing kartogenin (KGN) in response to 980 nm NIR [100]. These UCNPs were fabricated from Tm/Er-doped UCNPs. Following their synthesis, the UCNP was coated with mesoporous silica, and the silica surface underwent modification with photoisomerised azobenzene. Moreover, cell-targeting RGD molecules were affixed to the composite. The size of the UCNP was 55 nm, with the silica layer measuring 10 nm. Upon 980 nm NIR irradiation, the UCNP emitted UV light. Emission efficiency was assessed in relation to NIR intensity and irradiation time, reaching a maximum of 45%. Biocompatibility was measured by culturing the cells for 72 h in various concentrations of UCNP. The results after both 24 and 72 h indicated a survival rate of 90%, thus confirming that the UCNPs had no cytotoxic effects. After 7 days of differentiation at the optimised intensity and time of NIR irradiation, the expression levels of aggrecan, type II collagen (Col II), and SOX9 were compared between the differentiated cells and those subjected to UCNPs without NIR irradiation. The results showed significantly elevated gene expression in the group irradiated with NIR. Additionally, stem cell differentiation was monitored through nanofluorescence UCNPs. This study effectively demonstrated the suitability of synthesised UCNPs for promoting stem cell differentiation and monitoring, thereby validating their potential utility for both in vivo stem cell therapy and differentiation monitoring.

The conventional core-shell UCNP structure exhibits a relatively low upconverting yield. To overcome this limitation, the NIR intensity must be increased, or the irradiation time must be increased, which is accompanied by a change in temperature. In order to mitigate these temperature fluctuations, a core-shell-shell UCNP structure was engineered to enhance yield and promote NSC differentiation [101]. This innovative structure was composed of Tm, neodymium (Nd), and ytterbium (Yb). Moreover, the drug release was triggered at 808 nm NIR, which elicits minimal temperature change. NSCs were successfully differentiated by RA-releasing UCNPs. Next, the particles were coated with mesoporous silica to create a space for RA loading, followed by modification with a photoresponsive polymeric shell containing spiropyran groups. Afterward, Arg-Gly-Asp (RGD) was attached to the surface of target cells. The fabricated UCNPs were 144 nm in size. When irradiated with 808 nm NIR, spiropyran is converted to merocyanine, and the drug is released. The emission efficiency of the RA-loaded UCNPs was analysed by turning on and off the NIR source (power density 1.05 W/cm^2^) at intervals of 5 min, and the emission efficiency was 40% after 100 min of NIR exposure. Biocompatibility testing involved varying the laser power density for 5-min intervals, with results indicating increased cytotoxicity with higher densities. Across various UCNP concentrations, no cytotoxicity was observed up to 100 µg/mL over a 2-day culture period. Differentiation of NSCs occurred over 14 days with RA-loaded UCNPs. Notably, by day 5, expression of neuron-specific class III β-tubulin (TUJ1), an early neuronal differentiation marker, was 7-fold higher in the UCNP + NIR group compared to the control. On day 14, heightened expressions of neuron-specific microtubule-associated protein 2 and synapsin, indicative of synapse formation, were also evident. This research showcased the enhanced yield of core-shell UCNPs and their efficacy in fostering NSC differentiation, highlighting their potential for minimising errors during stem cell differentiation. These findings demonstrated the ability of UCNPs to selectively release differentiation factors through NIR irradiation while also allowing for the monitoring of stem cell differentiation. Therefore, UCNPs warrant additional research as functional candidates in the field of organoids, where the regulation of diverse differentiation factors is paramount.

## 7. Conclusions

### 7.1. Limitations

The present study provides a comprehensive overview of the recent advancements in the manipulation of stem cell differentiation using porous functional materials. The introduced porous functional materials demonstrated excellent performance in effectively stimulating stem cell differentiation. Materials based on mesoporous silica, including MSNs, have been utilised in combination with magnetic nanoparticles and UCNPs and have been demonstrated to effectively control stem cell differentiation through surface modifications. When integrated into scaffolds, PLGA has exhibited promising outcomes in facilitating stem cell differentiation. MOFs also exhibit outstanding capabilities in protecting and releasing unstable differentiation factors, while magnetic nanoparticles have been employed to enable not only cell differentiation but also additional functional integration. Furthermore, UCNPs have been harnessed for the targeted release of factors via near-infrared irradiation. 

### 7.2. Future Directions

As our understanding of stem cell differentiation advances, there is a growing interest in controlling stem cell behaviour through the use of various differentiation factors. To achieve the ultimate goals of stem cell-based technologies, such as drug screening, disease modelling, and stem cell therapies, controlling the behaviour of stem cells at a specific location and time is crucial. The use of drug delivery systems based on porous functional materials represents an effective means to modulate stem cell differentiation by controlling the behaviour of stem cells. Moreover, this approach can be extended to more complex stem cell differentiation processes, such as organoid differentiation.

## Figures and Tables

**Figure 1 jfb-14-00483-f001:**
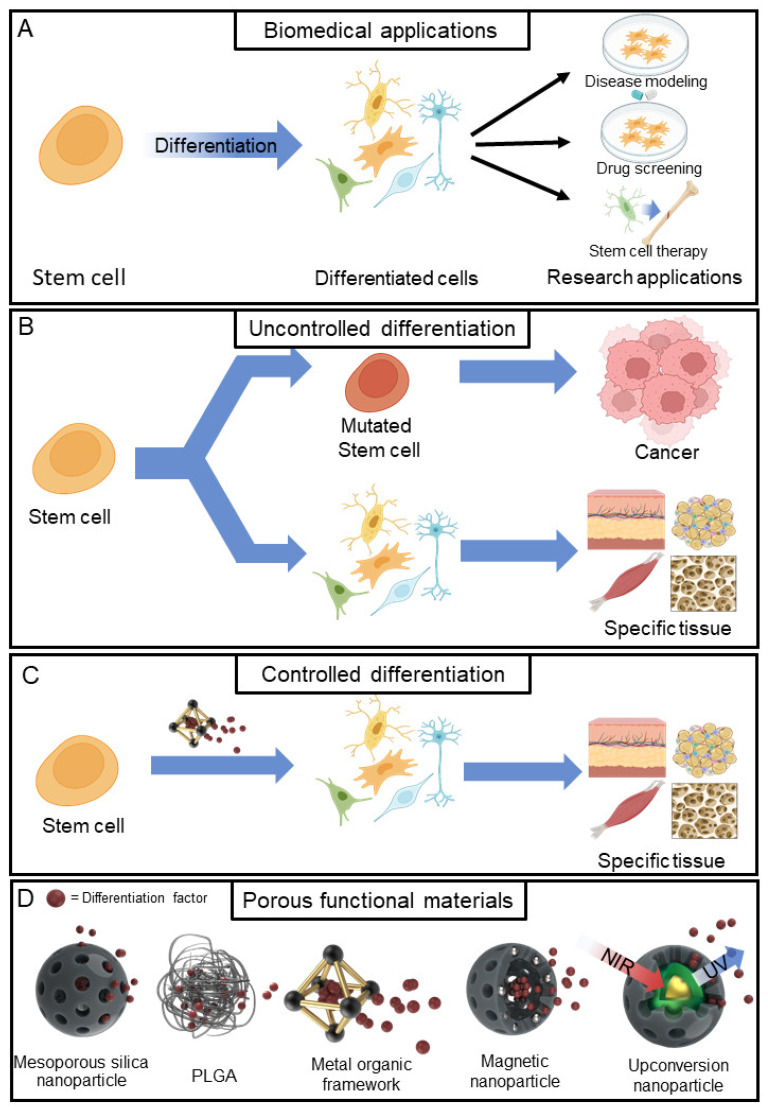
Graphical illustration of various applications of stem cell differentiation and comparison between uncontrolled and controlled differentiation and porous functional materials for differentiation. (**A**) Biomedical applications of stem cells. (**B**) Uncontrolled stem cell differentiation. (**C**) Controlled stem cell differentiation with porous functional materials. (**D**) Various porous functional materials are used for the fabrication of drug delivery systems. Created with BioRender.com (accessed on 11 September 2023).

**Figure 2 jfb-14-00483-f002:**
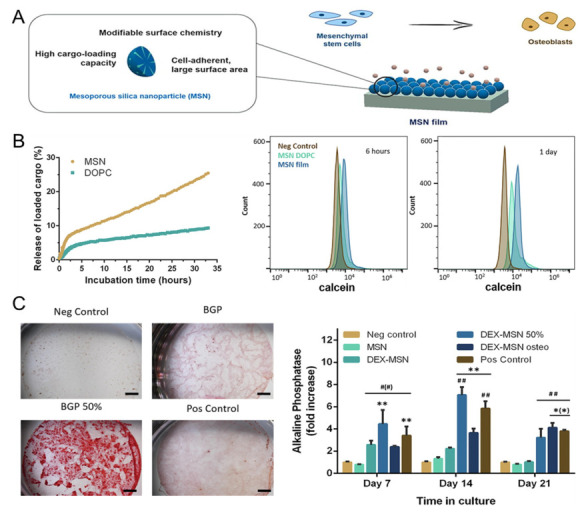
DEX-loaded MSN film for osteogenesis. (**A**) Schematic illustration of osteogenesis with MSN film. (**B**) Calcein’s release profile for MSN. (**C**) Comparison of osteogenesis progression between each group. Visualisation of calcium deposits (**left**) and ALP expression (**right**). *, significant difference compared to negative control; #, significant difference compared to MSN basic. * *p* < 0.05, ** *p* < 0.01, ## *p* < 0.01. With permission from [87]. Copyright 2019, Elsevier.

**Figure 3 jfb-14-00483-f003:**
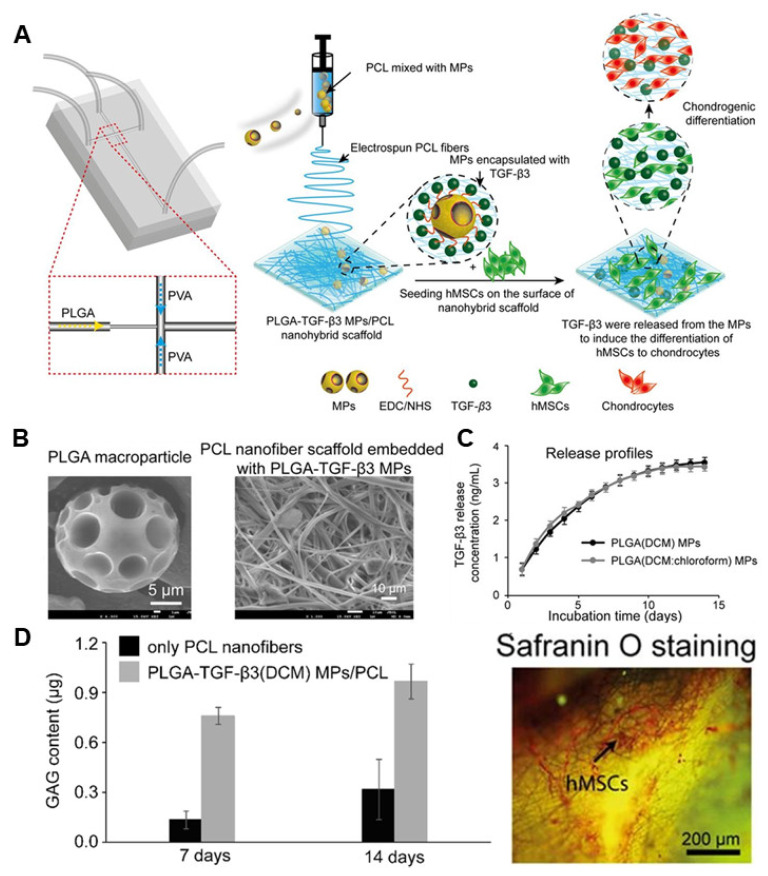
PLGA-based scaffold for chondrogenic differentiation. (**A**) Schematic illustration of the fabrication process of a PLGA-based scaffold. (**B**) Characterisation of PLGA macroparticles using SEM. (**C**) TGF-β3 release profile through the PLGA-based scaffold. (**D**) Evaluation of chondrogenic differentiation on a fabricated PLGA-based scaffold. With permission from [92]. Copyright 2020, Elsevier.

**Figure 4 jfb-14-00483-f004:**
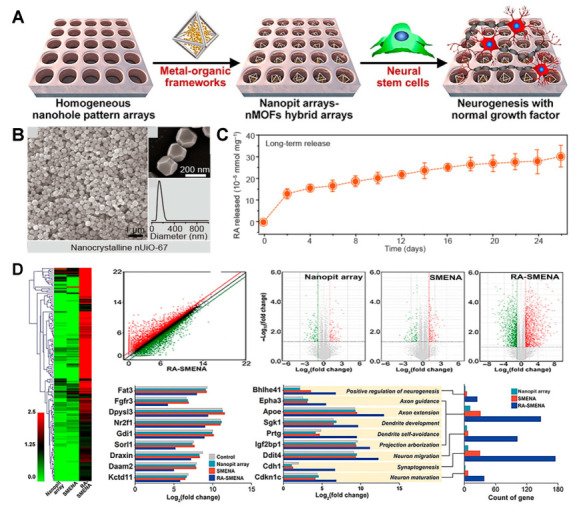
MOF-embedded nanohole array for NSC differentiation. (**A**) Schematic illustration of the MOF nanohole array. (**B**) Size characterisation of UiO-67 using FE-SEM imaging and dynamic light scattering (DLS) particle size distribution. (**C**) RA release profile. (**D**) Neurogenesis-related gene expression. With permission from [93]. Copyright 2022, Science.

**Figure 5 jfb-14-00483-f005:**
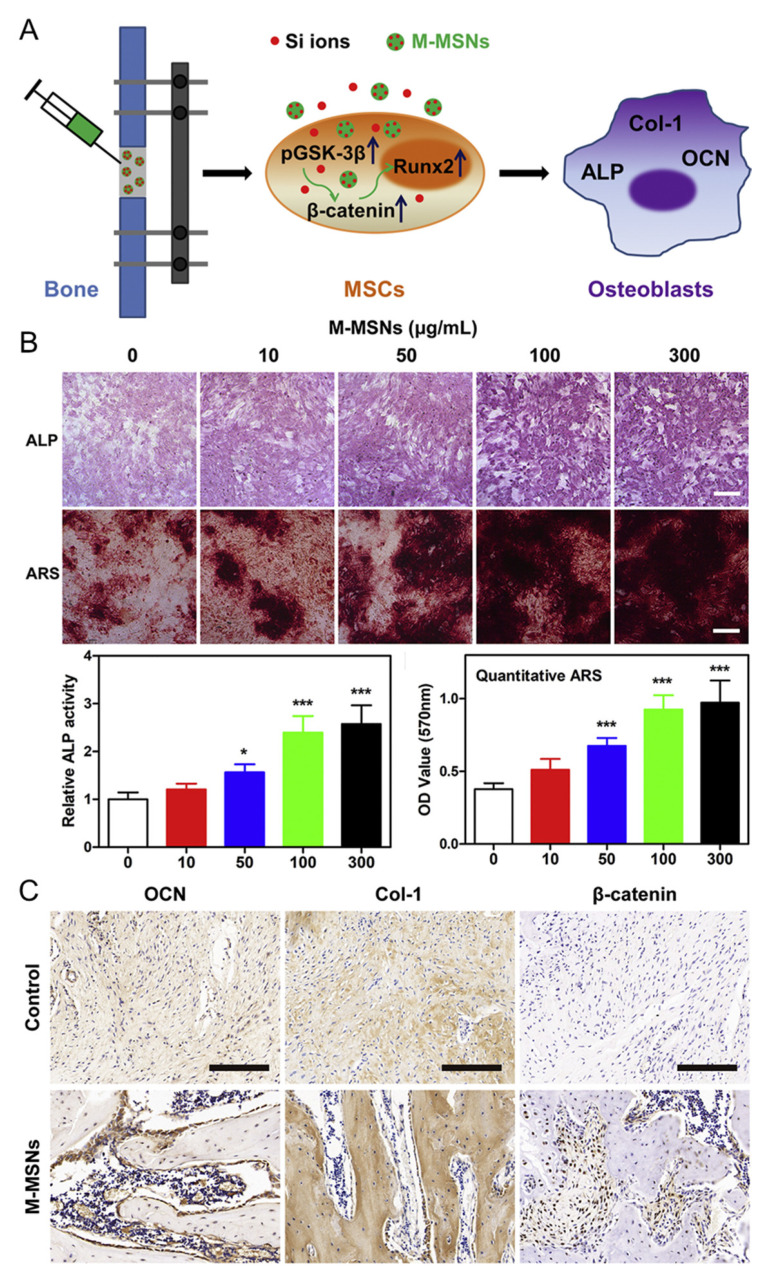
Application of magnetic nanoparticles in osteogenesis. (**A**) Schematic illustration of the osteogenesis stimulation process using magnetic nanoparticles. (**B**) Evaluation of osteogenesis enhancement by magnetic nanoparticles. (**C**) Immunohistochemical staining for the characterisation of osteogenesis. * *p* < 0.05, *** *p* < 0.001. With permission from [96]. Copyright 2019, Elsevier.

**Figure 6 jfb-14-00483-f006:**
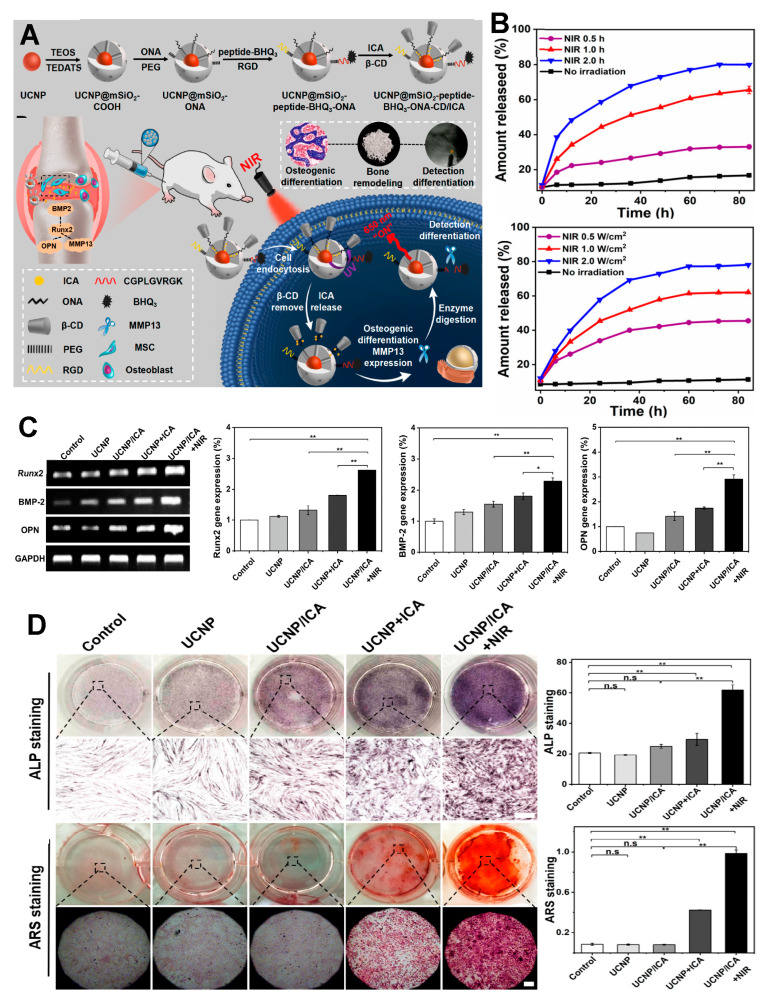
MSC osteogenesis with icariin-loaded UCNPs. (**A**) Schematic illustration of the ICA-loaded UCNP synthesis process. (**B**) Icariin release profile according to NIR irradiation. (**C**) Comparison of differential gene expression via RT-PCR. (**D**) ALP/ARS staining of the osteogenesis process. n.s, not significant. * *p* < 0.05, ** *p* < 0.01. With permission from [99]. Copyright 2022, ACS publications.

**Table 1 jfb-14-00483-t001:** Porous functional materials for stem cell differentiation.

Material	Target Stem Cell	Differentiation Factor	Differentiation Type	Ref.
Mesoporous silica nanoparticle (MSN)	hMSC	Dexamethasone (DEX)	Osteogenesis	[87]
Mesoporous silica nanoparticle (MSN)	bMSC	Dexamethasone (DEX)	Osteogenesis	[88]
Mesoporous silica nanoparticle (MSN)	rbMSC	Dexamethasone (DEX)	Osteogenesis	[89]
PLGA	hDPSC	TGF-β1	Chondrogenesis	[90]
PLGA	MSC	BMP-2	Osteogenesis	[91]
PLGA	hMSC	TGF-β3	Chondrogenesis	[92]
UiO-67	NSC	Retinoic acid (RA)	Neurogenesis	[93]
ZIF-8	MSC	Dexamethasone (DEX)	Osteogenesis	[94]
MIL-100(Fe)	NSC	siSOX9, Retinoic acid (RA)	Neurogenesis	[95]
Fe_3_O_4_, Mesoporous silica	MSC	Si ion	Osteogenesis	[96]
Fe_3_O_4_, Mesoporous silica	hBMSC	Minocycline	Osteogenesis	[97]
Fe_3_O_4_, graphene	BMSC	Nano-hydroxyapatite	Osteogenesis	[98]
Tm, Er (UCNP), Mesoporous silica	MSC	Icariin (ICA)	Osteogenesis	[99]
Tm, Er (UCNP), Mesoporous silica	MSC	Kartogenin (KGN)	Osteogenesis	[100]
Tm, Nd, Yb (UCNP), Mesoporous silica	NSC	Retinoic acid (RA)	Neurogenesis	[101]

## Data Availability

Not applicable.

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
