# Peer review of "Recent Advances in Stem Cell Differentiation Control Using Drug Delivery Systems Based on Porous Functional Materials"

_jfb, 2023, doi:10.3390/jfb14090483_

Round 1

Reviewer 1 Report

Specific Comments:

1. Title

Comment: It would be beneficial to the readers if the authors consider revising the title to convey the main topic of discussion i.e., on utilizing functional porous materials for controlled stem cell differentiation specifically through controlled release of differentiation factors. The current title implies that the manuscript focuses on how the porous functional materials themselves guide the differentiation of stem cells through their material properties which is not the focus of the presented discussion. Further, the abstract itself highlights that - “In this review, we discuss recent research cases that have effectively utilised porous functional materials-based drug delivery systems to regulate stem cell differentiation” (Lines 16-17).

2. Section 1; Introduction; Lines 38 - 40; “Another strategy involves creating an extracellular matrix (ECM) environment that matches the expression of integrin…………”

Comment: Authors should kindly consider rephrasing this sentence as it incorrectly implies that ECM contains integrins.

3. Section 1; Introduction; Lines 80 - 83; “This review summarises the recent advancements in stem cell differentiation control technologies using various types of porous functional materials that enable the regulation of stem cell differentiation…………”

Comment: Again, it would be beneficial to the reader if the authors specify utilizing functional porous materials for controlled stem cell differentiation specifically through controlled release of differentiation factors.

4. Section 2; Mesoporous silica nanoparticles; Lines 96 - 107

Comment: The authors should consider including a discussion on whether the study was able to (or unable to) decouple the effects of substrate properties and controlled release of differentiation factors on the stem cell differentiation process.

5. Figure 2C

Comment: What is the difference between DEX-MSN osteo and Pos Control?

6. Section 3; Polylactic-co-glycolide acid

Comment: The authors should consider including a discussion on whether the studies were able to (or unable to) decouple the effects of substrate properties and controlled release of differentiation factors on the stem cell differentiation process.

7. Section 6; Upconversion nanoparticles; Lines 433 - 436; “Additionally, successful in 433 vivo treatment was achieved by applying the developed UCNP……. highlighting its efficacy for osteoporosis treatment.”

Comment: It would be beneficial to the reader if the authors could include a brief discussion on how exactly the NIR modulation of UCNPs was carried out in vivo to highlight the in vivo translation of the technique.

8. Figure 6D

Comment: Figure 6D is not cited and discussed in the text.

9. Section 7; Conclusions

Comment: The conclusions need to be revised to stress on utilizing functional porous materials for controlled stem cell differentiation specifically through controlled release of differentiation factors.

 Minor comments:

1. References are missing on Lines 74, 162, and 183 respectively.

2. Table 1

Comment: It would be beneficial to the reader if the authors could include a footnote containing a list of abbreviations in the table.

3. At several places in the manuscript, the beta sign is incorrectly printed when referring to TGF-beta.  

4. Authors need to re-edit the manuscript to address minor grammatical errors to improve its readability. 

The article is generally well-written and well-articulated, however, minor grammatical errors need to be corrected to improve its readability.

Reviewer 2 Report

The authors reviewed 'Recent advances in porous functional materials to control stem cell differentiation'. The manuscript is well-written. Only some minor suggestions should be made. 

Figure1 B. The figure should be redesigned, it is explained in the text the stem cells need some mutations and transformation to turn into cancer cells but the figure shows a direct way. Should be changed. 

Line 50-52: 'Moreover, high-concentration media containing differentiation factors, such as retinoic acid, dexamethasone, and glucocorticoids, have been demonstrated to induce cellular stress and trigger apoptosis in stem cells' The sentence should be rewritten to make sure the readers understand that the high concentration of these factors create apoptosis. It is not clear. 

Line 72: 'Additionally, they exhibit biocompatibility and protect drugs from degradation [71-73].' The sentence is not clear and may be misleading. The material shows biocompatibility but in this way sentence says that the porous materials are biocompatible. It should be rewritten.

Line 118: Standard deviation should be supplied, if possible.

Line 170: Should be 3 Dimensional (3D)

Line 241: Standard deviation should be supplied, if possible.

Lİne 284: Please do not use 'a recent study'.

Line 341: The following sentence should be rewritten 'Therefore, these results suggest that magnetic nanoparticles can promote osteogenesis in vivo and in clinical applications.'

Reviewer 3 Report

This is a well-written review on a subject that has seen many similar reviews but benefits from updated versions that can incorporate the most recent advances. The images are detailed and well-prepared. The writing style is appropriate for a review and I have no complaints about the materials covered. Missing references can always be identified in review papers, but in my opinion, the list included is comprehensive and representative of the studies published. This can be very useful for researchers interested in embarking in new projects related to the area covered here.

Reviewer 4 Report

Dear Editor,

This is an interesting title and authors reviewed recent research cases that have effectively utilised porous functional materials-based drug delivery systems to regulate stem cell differentiation. My comments are listed below:

- Abstract; aim should be explain after first sentence, method should be explain, authors should explain their conclusions in this review article

-Authors should explain about the epigenetic in stem cell differentiation, it would be better to use the below reference;

The Role of Epigenetic in Dental and Oral Regenerative Medicine by Different Types of Dental Stem Cells: A Comprehensive Overview, Journal: Stem Cells International 2022-06-09 | Journal article DOI: 10.1155/2022/5304860 CONTRIBUTORS: Ahmed Hussain; Hamid Tebyaniyan; Danial Khayatan

-It would be better to add a title as limitations or explain the limitations in conclusion

-It would be better to add a title as future direction after conclusion section

Best, 

Round 2

Reviewer 4 Report

Dear Editor,

The revised version is acceptable.

Best wishes,